# Impact of Climate Change on Rural Poverty Vulnerability from an Income Source Perspective: A Study Based on CHIPS2013 and County-Level Temperature Data in China

**DOI:** 10.3390/ijerph19063328

**Published:** 2022-03-11

**Authors:** Qihang Li, Peng Sun, Bo Li, Muhammad Mohiuddin

**Affiliations:** 1Center for Economic Research, Shandong University of Finance and Economics, Jinan 250014, China; lqh_sdufe@163.com; 2School of Economics, Shandong University of Finance and Economics, Jinan 250014, China; ytsunpeng@163.com; 3School of Management, Tianjin University of Technology, Tianjin 300384, China; 4Faculty of Business Administration, Laval University, Quebec, QC G1V 0A6, Canada; muhammad.mohiuddin@fsa.ulaval.ca

**Keywords:** poverty vulnerability, climate, generalized three-stage least squares method, threshold regression

## Abstract

Harsh natural climatic environments, such as extreme weather and natural disasters, cause devastating blows to production activities and increase the probability of geographic poverty, climate poverty, and return to poverty. Thus, this study uses climate data and micro survey data (CHIPS2013) to examine the impact of climate on vulnerability to individual poverty in rural China. The results demonstrated that extreme temperatures (hotter summers, colder winters, and greater day-to-day temperature gaps) reduce vulnerability to poverty. This was also supported by the median and average temperatures. Second, there is an association between poverty vulnerability and poverty; that is, poorer people will become poorer with an increase in poverty vulnerability. In fact, in the case of higher income, the higher the probability of returning to poverty, the higher the vulnerability. Policy formulation processes should take into consideration different types of impacts from harsh climate on different vulnerable groups. No single action might be adequate and an integrative approach integrating various strategies and actions are required to overcome challenges posed by climate change and poverty vulnerabilities.

## 1. Introduction

Global rise in temperature has become an indisputable fact. Hansen et al. [1] had predicted that global temperature would continue to rise in the 21st century and that this phenomenon originated from human activities. Climate has a particularly negative impact on developing countries, rural areas, and on agricultural production. Authors [2,3] found that climate change affects the agricultural production of the poorest and most vulnerable people in the tropics. Examining the relationship between climate change, agricultural sustainability, and poverty is complex [4,5].

China has a vast territory and diverse types of terrain, with almost all climate zones, such as continental nature, plateau and mountain areas, monsoon climate, and even rainforest climate, which leads to frequent natural disasters. According to the Université Catholique de Louvain (UCL), the database on international natural disaster shows that a total of 1098 natural disasters occurred in China between 1990 and 2020, including 15 extreme temperatures with an increasing trend year by year. Among them, the worst was the snow disaster in 2008, which affected 19 provincial administrative regions. Due to low temperature, rain and snow, and freezing disasters, crops were affected by more than 7000 hectares, and the total direct economic losses reached more than 50 billion yuan (RMB price in 2008).

At the same time, China is an important agricultural country and the largest grain producer, accounting for about 8% of the world’s arable land. According to China’s 2020 census data, more than 500 million people live in rural areas in China, accounting for 36.11% of the total population, and a relatively large number of people are still engaged in agricultural production activities. However, at the same time, China’s per capita cultivated land area is only 1.3 mu, which is 3.5 mu less than the world average. The fine fragmentation of cultivated land with households as the main body weakened their ability to resist natural and man-made disasters. Extreme temperature and drought would seriously affect grain production [6,7]. 

China achieved huge success with its poverty alleviation strategies. However, at the same time, it is not guaranteed to consolidate the achievements of poverty alleviation. There are weak agricultural and industrial foundations, homogenization of industrial projects, and environmental vulnerability to economic and social arenas. There is still a risk of returning to poverty among the people who have been already lifted out of poverty, and there is still a risk of causing poverty among the marginal population. World Bank report [8] predicted that, by 2030, climate change could put 32 to 132 million people worldwide into extreme poverty. The health effects of climate change and the effects of food prices are the main causes of extreme poverty. Given the population size, importance of agriculture and industrial sectors in national economy, and the vulnerability of the rural population in China, studying the impact of climate change issues on rural poverty is a very important topic.

In terms of poverty measures, Klasen and Waibel [9] mentioned that both developed and developing countries should measure poverty population (head count poverty), poverty gap (poverty gap), and poverty severity (poverty severity). Energy poverty is different from poverty vulnerability. The World Bank defines poverty vulnerability as the probability of future poverty. Falling crop harvest, higher food prices, and major household labor diseases may all increase the vulnerability to poverty. The concept of poverty vulnerability has been introduced into the definition of poverty. In addition to the low income-based basic social welfare indicators, poverty should also include poverty vulnerability caused by external shocks [8]. However, energy poverty is mainly reflected in the low living energy use level, poor energy use structure, weak energy use capacity, and the resulting health and social and economic consequences. Energy poverty is widely found in developing countries and regions, including China, and is a great development concern for the United Nations, the International Energy Agency, and other international energy organizations. Therefore, this study examines the impact on the vulnerability of rural poverty in China from the novel perspective of the external impact of extreme temperature. The study uses weather data to investigate the impact of climate on vulnerability to poverty in rural areas. In doing so, the study aims to provide relevant policy suggestions on optimizing poverty alleviation programs in rural areas in developing countries, along with an exploratory approach to prevent rural villagers from returning back to poverty. 

This study examines the impact of climate on the vulnerability of individual poverty in rural China, using climate data and micro-research data (CHIPS 2013). The benchmark results found that extreme temperatures (hotter summer, colder winter, and greater daily temperature difference) help reduce poverty vulnerability. The same conclusion was found after using the temperature median and mean.

After heterogeneous grouping, it is concluded that vulnerable people are more likely to fall into poverty in the face of high temperature, which can increase the awareness to restrain the non-vulnerable people. A colder winter is beneficial for both vulnerable and non-vulnerable populations. The higher altitude is conducive to reducing the probability of individuals returning to poverty, and migrant work behavior and entrepreneurship has the opposite effect on alleviating the vulnerability of poverty.

To further examine the sensitivity of individuals and return to poverty and temperature, differential responses to climate were found among different groups after the use of threshold regression, with a single threshold. The more poor and vulnerable people, the hotter the summer, and the colder temperature difference is likely to cause return to poverty, which also deepens and expands our basic conclusion. Moreover, there is a linkage effect between poverty vulnerability and poverty; that is, the poor people will be poorer with the increase of poverty vulnerability, and the higher the probability of a return to poverty, the more vulnerable people are. Finally, individuals are added into county-level cities and results show that different provinces should adopt different targeted strategies.

The main contributions of this paper are as follows: (i) This study is likely to enrich country studies on climate change and poverty. On the one hand, several studies focus on the impact of floods, extreme temperatures, and extreme drought on the consumption of poor groups [10,11]. On the other hand, several others have tested the impact of drought, flood, irregular rainfall, high temperature, and strong wind on the vulnerability of poverty [12,13]. Study on China, a large agricultural country with frequent natural disasters, can enrich us to find ways of mitigating the climate change effects and use that knowledge for other vulnerable countries where climate change affects poverty. (ii) Based on Anton’s [14] (2021) idea of temperature on micro-subjects, this paper conducts an in-depth study of the differential effects of different temperatures on poverty vulnerability and fulfills the gap in the literature by undertaking an in-depth study both on the macro and micro level impact of climate change on different levels of poverty vulnerabilities.

The rest of the paper is arranged as follows: the second section includes literature review, data variables and descriptions, the third section describes the empirical analysis, and the final section concludes the paper along with its policy implications.

## 2. Literature Review

Extreme weather, a short-term manifestation of climate, threatens the basic survival and production of human beings. On one hand, Stone et al. [15] found that extreme heat—which is related to the continuous expansion of cities and the rise of surface temperatures in urban areas—causes more deaths each year than other climate-related extreme weather. Although extremely low and high temperatures increase mortality, when the coldest weather warms to a certain threshold, mortality begins to decrease. This phenomenon varies with latitude. Lower temperatures in the south correspond with higher temperature in the north, which causes an increase in the death rate [16]. In contrast, drought and high temperatures significantly reduced global crop yields from 1964 to 2007, but floods and extremely low temperatures did not have a significant impact on agriculture [5] In contrast, Barlow et al. [7] found that extremely high and low temperatures can significantly affect the harvest of food producers. 

Looking back at history, the relationship between human development and climate can be divided into three stages. Initially, in agricultural societies with low productivity, there was less interaction between humans and the climate. The expanding civilization and exploitation of the natural system did inflict negatively on climate change. It has affected agricultural production, health threats, and conflicts in preindustrial societies. The hot and humid summers brought prosperity to Rome and the Middle Ages, while climate change coincided with the destruction of the Western Roman Empire and the Great Migration riots [17]. In Western Europe, longer summers have facilitated the production of bumper crops and population growth, allowing culture to flourish. However, in other parts of the world, global warming has caused droughts and famine [18]. Furthermore, the interaction between humans, weather, and climate has increased. At first, settlers who migrated from Britain to North America believed that climate was the same at all latitudes. After discovering the inability of a hot colonial climate to produce rich products, they began collecting information on climate patterns, growing seasons, and different crops [19]. 

Nowadays, people are motivated to control climate, which is against the laws of nature. Authors [20,21] believed that traditional attempts have focused on controlling and mastering future climate without exception, which reflects arrogance and utopianism. Therefore, we need to pay attention to human subjective initiative and realize the harmonious coexistence between humans and nature while respecting objective laws. 

Further, Luber et al. [22] used a generalized climate cycle model and determined that the frequency and intensity of high temperatures will continue to increase. This is particularly evident in high-latitude regions, where metropolitan populations are not able to adapt to climate change. Exposure to extreme heat not only increases mortality rates but also causes more prominent public health problems. Similarly, Rosenzweig [23] used the GCM model to predict that high latitudes and altitudes will become warmer, especially in winter, and warm winters will cause an increase in the number of pests and pathogens. However, climate change has been beneficial for some regions, at least in the short term. According to Anton [14], higher temperatures increase the profitability of energy and gas-related firms. This might be the case for oil and gas firms drilling in cold weather in the northern countries, but might not have the similar effects on agriculture and industrial firms in the warmer environment in the southern hemisphere. In fact, developing countries face the threat of reduced food production and increased malnutrition. 

Therefore, from a long-term perspective, exploring human survival and development from a climate perspective has great theoretical significance and practical value. Vulnerability to poverty—an important measure of poverty—signifies the probability of not being in poverty now but facing poverty in the future. Further, those who are currently experiencing poverty will continue to do so in the future as well. Although Dutta et al. [24] believed that there is no consensus on the link between poverty and vulnerability, they estimated the distribution of the future expenditure of each household and used vulnerability to calculate the distribution of these distribution functions. This determines that vulnerability to poverty can reflect individual poverty more accurately, to a certain extent. 

Numerous studies focus on climate change and poverty vulnerability in African regions. Samuels et al. [13] test the effects of drought, heat, and strong winds on poverty vulnerability in the indigenous Nama community in South Africa. Maganga et al. [12] examined the effects of drought, flood, and irregular rainfall on the poverty vulnerability of small farmers were investigated using survey data from Malawi. Ahmed et al. [25] studied the impact of climate fluctuations on the vulnerability in the province of poverty in Tanzania, which was predicted in the late 20th century using climate prediction models, statistical crop models, and general equilibrium simulations. Azzarri and Signorelli [10] used large data from a micro survey of 24 sub-Saharan countries and found that although floods will significantly reduce total (per capita) food consumption and increase extreme poverty groups, the effects of extreme temperatures and extreme drought are indeed uncertain. Finally, Marco et al. [11] adopted Tanzania panel data to find that the impact of extreme temperature has a significant negative impact on rural household consumption.

In the study of poverty vulnerability from a Chinese macro perspective, some scholars have analyzed the effectiveness of poverty reduction policies in reducing the vulnerability of farmers to policies, such as mutual aid funds for poor villages, farm cooperation insurance, and urban and rural minimum living security [26,27,28,29,30,31,32]. According to the existing results, this policy did not play a substantial role and had no obvious effect on low-income families and vulnerable groups, which may be related to “elite capture” and “targeting bias” in rural financial loans. However, Zhang and Yin [33] believed that financial services play an important role in reducing vulnerability to poverty. Simultaneously, for income redistribution links, such as fiscal policy, public transfer payments have no impact on the vulnerability to chronic poverty and temporary poverty [34]. The ineffectiveness of such policies, especially social policies, may stem from their exclusionary nature; these constitute long-term rural poverty [35]. Other scholars have found that trade openness can significantly reduce rural households’ vulnerability in China [36]. 

At the micro level, from the community perspective, poverty prevention effects on the comprehensive development of participatory communities is significant, but there is no significant long-term time lag effect [37]. From the individual household perspective and household endowment, the age and livelihood fragility of farmers have an inverted “U” association [38]. Human capital characteristics are closely related to the probability of returning to poverty [39]. Not only do individual or family endowment factors affect the probability of returning to poverty, shock-events are among the main influencing factors [40]. The elderly population’s vulnerability increases in the face of risks [41]. In addition, from the perspective of individual farmer behavior, labor migration can significantly reduce vulnerability to poverty [30,31], while farmers’ self-employment can significantly reduce the vulnerability of non-poor families [42]. 

Scholars have combined various indicators to explain the effectiveness of various forms of capital. Huang [43] demonstrated the effectiveness of physical and social capital. In the context of the aforementioned statement, Chen [44] believed that the best way to reduce vulnerability to poverty is human capital, followed by natural capital, physical capital, social capital, and financial capital. 

Geographically speaking, the western region is more ecologically fragile and the environment is harsher. For instance, the vulnerability of farmers in the Qinba Mountains area is significantly higher than the incidence of poverty because of the impact of resource endowment [45]. Wang et al. [46] believed that the crux of rural poverty in the west lies in the vulnerability of farmers toward multiple risks. Based on this view, Wan et al. [47] proposed the importance of relocation and believed that Chinese farmers should accumulate more productive material capital, human capital, financial capital, and social capital while increasing the efficiency of asset use to reduce vulnerability to poverty. Furthermore, in a research review of the impact of the environment and disasters on poverty, Cheng et al. [48] summarized that the poor need a variety of poverty alleviation models. Current research rarely considers the environmental and disaster factors. Similarly, Zhang et al. [49] considered the important role of poverty alleviation under climate change to promote the development of related research. 

A review of the extant literature suggests that existing poverty alleviation policies are not effective to a certain extent and the endowments of farmers and their families, such as human, physical, social, and financial capital, cannot be changed in the short term. In addition, shocking events increase the probability of individuals and households returning to poverty. More importantly, existing poverty alleviation work focuses on financial support and skills training instead of focusing on the impact of extreme weather on individuals and families returning to poverty. This is especially relevant for the livelihoods of people in remote mountainous and ecologically fragile areas. Therefore, in the context of winning the fight against precision poverty alleviation, it is particularly important to examine the vulnerability of poverty from a climate perspective and to provide constructive suggestions for the development of rural areas in other developing countries, such as China. 

The research of this paper has certain theoretical and practical significance, mainly reflected in the following aspects: (1)Theoretical significance: (i) It can be predicted that climate change and global epidemics will pose major challenges for the return to poverty in the future. Thus, studying these external shocks will further enrich the World Bank’s definition of poverty vulnerability [8]. (ii) There is a wealth of studies on climate change and poverty focusing on African countries [10,11,12], such as Tanzania, South Africa, Malawi, and in 24 sub-Saharan countries. This also coincides with the greater impact of climate change on the southern Sahara region. China is a big agricultural country with frequent natural disasters, so research on China can enrich the diversity of countries that climate change effects on poverty.(2)Practical significance: (i) This paper provides experience for the developing countries focusing on agricultural production. (ii) Due to the different latitude, longitude, and temperature zone of different countries, the conclusion that summer is hotter and colder in winter and the greater daily temperature difference based on the micro-survey data of rural China applies to different countries.

## 3. Data and Variables

### 3.1. Data Selection

For climatological data, the CNDRS database was used to collect data on 365 days of high and low temperatures at the county level in 2017, as well as the longitude, latitude, and altitude of county-level cities. The data used in this study were obtained from the China Household Income Survey (CHIPS). Data were provided by the China Income Distribution Institute of Beijing Normal University. The survey began in 1989. It primarily investigates data on Chinese household income and supports the study of Chinese household income and expenditure. The CHIP2013 data was the fifth round of a nationwide survey conducted by Chinese residents’ income projects in July–August 2013, which mainly collected income and expenditure information for the entire year of 2012. The sample covered 18,948 household samples and 64,777 individual samples drawn from 234 counties and districts in 126 cities in 15 provinces. This includes 7175 urban household, 1013 rural household, and 760 outdoor migrant samples. The CHIP2013 data include related information on farmers’ business and industry, financing channels, and individual and family characteristics of farmers, assets, and income. 

The original samples were cleaned and sorted to eliminate bias in the original statistical information. The following measures were taken: (1) Samples with abnormal values in the micro-control variables were excluded; (2) County-level city samples that did not correspond to the macro variables in the county were assessed; and (3) County-level cities that did not correspond to altitude, latitude, or longitude were assessed. After the treatment, we obtained 13,218 individual samples from rural residents. The sample included 11 provinces (municipalities and autonomous regions) in China, of which 336 are in Shanxi, 842 in Liaoning, 1383 in Jiangsu, 1383 in Anhui, 1371 in Shandong, 2035 in Henan, 1160 in Hubei, and 1666 in Hunan. Further, 1821 samples were obtained from Guangdong Province, 977 samples from Sichuan Province, and 667 samples from Gansu Province. As the samples from all provinces and cities have a certain volume, they are representative. From the perspective of the east, middle, and west distribution, which is based on the 3441 standard line and grouping according to a 29% vulnerability probability, it was found that approximately 33%, 42%, and 45% of the samples in the eastern region, central region, and western region, respectively, were in the vulnerable group. This is in line with the fact that the eastern part is developed, central part is underdeveloped, and western part is backward. Vulnerability is distributed stepwise in the east, middle, and west. 

### 3.2. Variable Description

The five poverty vulnerability standards measured using the VEP method, i.e., *poor_$1*, *poor_low*, *poor_$1.5*, *poor_high*, and *poor_$2* were the explanatory variables.

We use the World Bank’s 1 US dollar/day, 1.5 US dollars/day, and 2 US dollars/day income as the poverty line standards that researchers are accustomed to adopt, combined with the purchasing power parity (PPP) exchange rate and the CPI adjustment of urban and rural living costs in different regions provided by the database to obtain the international standard measured in RMB Standard poverty line. As the determination of the threshold value of vulnerability is subjective and arbitrary, this paper uses two threshold values of relative poverty vulnerability for robustness analysis: 30% of the median rural per capita disposable income of the National Bureau of Statistics in 2013 is used as the vulnerability. The first threshold of vulnerability, namely low vulnerability, takes 50% of the median rural per capita disposable income of the National Bureau of Statistics in 2013 as the second threshold of vulnerability, that is, high vulnerability. The five thresholds have been presented at the Table 1 as follows:

On the one hand, the vulnerability to individual poverty in the sample population will gradually increase with the increase in the standard line. Through this, we can observe the monotonic transformation of the coefficients under different standard lines. On the other hand, although the five criteria of classification are quite different, they can be used as mutually supportive evidence to substantiate that climate impacts poverty vulnerability. 

Some scholars believe that the long-term change cycle and stable nature of the climate lasts as long as 30 years. Due to data restrictions, we could only obtain the annual average temperature values of provincial capitals over the past 30 years (from 1990 to 2018). As shown in Figure 1, the average annual temperature was roughly divided into three levels in the 11 provinces (−20–25 degrees, 20–15 degrees, and below 10 degrees). It has not changed much in the past 30 years, which confirms that temperature is an important proxy variable for climate. It also provides support for problems caused by data restrictions. 

Therefore, the proxy variables for measuring the climate, that is, the maximum value of high temperature, minimum value of low temperature, and maximum value of daily temperature gap in the temperature of each county and city in 2013, were used as core explanatory variables. The median and average values of high and low temperatures were added in the robustness analysis. 

### 3.3. Control Variables and Endogenous Treatment

First, as climate is an exogenous macro variable and individual poverty vulnerability is a manifestation of micro-individual behavior, there is no two-way cause-and-effect problem. Second, the VEP method was used to measure individual vulnerability to poverty. There was a strong correlation between vulnerability and personal traits. 

To alleviate the error of the unobservable variables, we used the current income of the individual as the control variable in the first stage of the VEP measurement as the future income forecast. Second, we added several individual trait variables, such as ethnic variables that measure individual cultural habits; hukou, party, and village administrator variables that affect the endowment and acquisition of personal factors; health variables that affect individuals’ normal work, study, and life; medical care for personal health protection insurance variables; and number of years of education that affects individual acquisition of new knowledge and skills. As our sample is scattered among 11 county-level cities, we have added the longitude and latitude of the county-level cities as macro-control variables, considering that the north-south and east-west differences are large. 

In addition, the CNDRS county-level data collection period was relatively short. Long-term weather conditions could not be obtained for an area. Therefore, based on the cross-sectional micro data analysis, they cannot automatically eliminate characteristics that change with time but do not change with individuals, such as cultural habits. However, as climate is a region’s long-term, basically steady and observable variable, we conducted an empirical analysis based on existing weather data and individual micro data. Simultaneously, we added county-level fixed effects, considering the inherent characteristics that do not change with county. Further, we used clustering robust standard error at the county level, which strengthened the persuasiveness of our causal effect explanation. Finally, measurement of micro data and the problem of errors has already been discussed before; therefore, it will not be repeated. In summary, we believe that the model has strong explanatory power for causal effects based on the interpretation and treatment of endogenous problems.

Table 2 provides a description of the variables. First, with an increase in the poverty and vulnerability standard line, the mean value of the probability of individual poverty gradually increases from 0.023 to 0.056. Second, at the temperature level, the median high and low temperatures and the average high and low temperatures are above 0 degrees. The former is larger than the latter. In addition, the maximum and minimum temperatures of different counties were not comparable. Consequently, we introduce the concept of annual maximum value of daily temperature gap to measure the difference between the highest and lowest temperatures of a county-level city in a year, wherein the maximum and minimum differences reached 26 and 14 degrees, respectively. Third, there is large individual heterogeneity among controlling variables at the individual level, such as ethnicity, party, village cadres, household registration, minimum consumption level, personal income, education level, health status, and medical security.

Finally, for other geographic factors, such as longitude and latitude, most provinces are located north of the Tropic of Cancer and there are more in the southern provinces. The lowest is 7 m above sea level in the plains, and highest is 1496 m in mountainous terrain. The average value is approximately 200 m, indicating that the terrain is hilly.

## 4. Empirical Research

### 4.1. Descriptive Test

First, we used the following five scales: *Poor_$1*, *Poor_low*, *Poor_$1.5*, *Poor_high*, and *Poor_$2*. Second, referring to the existing literature, the probability of poverty and fragility is grouped according to a 29% probability based on different levels of standard lines, for example the 2270 ($1) line. If the probability of personal poverty vulnerability was greater than 0.29, the individual was considered to be in the vulnerable group. Third, by calculating the sample’s vulnerability according to the five different standard lines, we obtained 4938, 5027, 5133, 5331, and 5356 vulnerable individuals among 13,218 sample observations. Thus, as the vulnerability line increases, an increasing number of people are incorporated into the vulnerable group from non-vulnerable groups. 

Finally, vulnerable populations were selected under different standards according to county geographical features (high temperature, low temperature, daily maximum temperature gap, and median altitude). See descriptive statistics in Table 3. We found that these geographical features are heterogeneous among the vulnerability to poverty group. First, people who are more vulnerable are located in high-altitude mountain plateau areas, which may be related to rugged mountain roads, poor infrastructure, and poor communication with external markets. Second, regions with larger temperature differences are conducive to sugar accumulation, higher yields, and more varieties of crops to be cultivated. This reduces the probability of vulnerability. Finally, the higher the temperature, the lower the probability of vulnerability.

As this is only a simple median temperature grouping, in the subsequent empirical analysis, we introduced the concept of extreme temperature and divided it into annual maximum temperatures (as counties and cities have a monsoon or continental climate, it is considered as the highest temperature in summer) and the lowest annual temperature (similarly regarded as the lowest temperature in winter). Simultaneously, the annual median and mean temperatures were introduced as a robustness check to verify our basic empirical results. In the mechanism analysis, we introduced geographical factors to perform heterogeneous grouping. 

### 4.2. Measurement of Poverty Vulnerability (VEP)

Vulnerability is defined as an adverse impact on welfare rather than exposure to poverty. Our definition includes vulnerable groups who are currently poor and likely to remain poor even if they have not suffered any major adverse welfare shocks. On the other hand, this definition excludes non-poor families who are currently wealthy enough to face large adverse shocks and will not face poverty if they are hit with such events. 

Formally, the vulnerability level of individual P at time t is defined as the probability that the family finds itself with poor consumption at time t + 1. Individual consumption at any time usually depends on several factors. These include wealth, current income, expectations of future income (i.e., lifetime outlook), uncertainty about future income, and spending power in the face of various income shocks. Each of these depend on various household and personal characteristics and may also embody some unobservable characteristics of the individual and the overall environment (macroeconomic and sociopolitical) in which the individual and family are located [49,50]. 

We use the following model to construct the vulnerability to poverty measured by the VEP method:

Formally, the vulnerability of individual *P* at time *T* can be expressed as:$$V_{PT}=\mathrm{Pr}\left(ln\_inc_{P,T+1}\right)\leq Z\;\;\;\;\;\;\;\;\;\;\;\;\;\;\;\;\;\;\;\;\;\;\;\;\;\;ln\_inc_{P,T}=C\left(X_{P},\beta_{T},\alpha_{P},e_{P,T}\right)$$
$$V_{PT}=\mathrm{Pr}(ln\_inc_{P,T+1})=C\left(X_{P},\beta_{T},\alpha_{P},e_{P,T}\right)\leq Z|X_{P},\beta_{T},\alpha_{P},e_{P,T})$$

The main idea of the VEP method is to use a three-stage generalized least squares estimation (FGLS) to first establish a model of income mean and income fluctuation, estimate the logarithm of per capita income, and perform OLS regression on the squared residuals after regression.
$$ln\_inc_{P,T}=X_{P}\beta +e_{p}$$
here, *ln_inc* represents an individual’s income level, and *X* represents individual characteristics, family characteristics, and some factors that are vulnerable to risk shocks. Thus, various factors have different degrees of influence on temporary and persistent income.
$$\sigma_{e,P}^{2}=X_{P}\theta$$

Based on the above regression, we constructed the heteroscedastic structure weights and re-weighted the regression of the residual squared and the income logarithm to obtain the estimated value.
$$\hat{E}[ln\_inc_{P}|X_{p}]=X_{h}\hat{\beta }\;\;\;\;\;\;\;\;\;\;\;\;\hat{V}=\mathrm{Pr}(ln\_inc_{P}<\ln Z|X_{P})$$

Therefore, vulnerability to poverty signifies the probability that an individual will face poverty in the future.
$$V_{P}=\hat{Pr}\left(ln\_inc_{P}\langle \ln Z|X_{P}\right)=\Phi \left(\frac{\ln Z-X_{h}\hat{\beta }}{\sqrt{h_{X}\hat{\theta }}}\right)$$

### 4.3. Benchmark Regression

Model introduction: *Poorstandard* included five standard lines: *Poor_$1*, *Poor_low*, *Poor_$1.5*, *Poor_high*, *Poor_$2*. *Max, min*, and *maxgap* are the main explanatory variables that represent the highest temperature, lowest temperature, and daily temperature gap, respectively. *X* is a series of control variables, which include both individual characteristic variables and latitude and longitude (macro variables of geographical features). Simultaneously, we controlled for county fixed effects.
$$Poor\_standard_{i}=max_{i}+\sum X_{i}+e_{i}$$
$$Poor\_standard_{i}=min_{i}+\sum X_{i}+e_{i}$$
$$Poor\_standard_{i}=maxgap_{i}+\sum X_{i}+e_{i}$$

From Table 4, we can see that high temperatures have a significant impact on reducing vulnerability to individual poverty. That is, the hotter the summer, the least the likelihood of an individual to return to poverty in the future. With the increase in the poverty and vulnerability standard line, that is, the increase in the average value of the group’s vulnerability probability, the effect of high temperature on reducing the vulnerability to individual poverty becomes increasingly obvious. The coefficient increased from −0.0131 to −0.0519. On the one hand, high temperature is conducive to the condensation of water vapors that facilitates precipitation. This, in turn, is more conducive for agricultural irrigation because it promotes agricultural production and income. On the other hand, high temperature is more likely to make individuals full of passion and with a willingness to work, reducing leisure. The more time people spend working, the more income they can earn, which reduces the vulnerability of individuals to poverty.

From Table 5, we can see that the higher the temperature, the worse the vulnerability of individuals to poverty. In other words, the colder the winter, the lower the probability of individuals to return to poverty in the future. With the increase in poverty and vulnerability standards, the lower the temperature and the stronger the ability to reduce the vulnerability of individuals to poverty. The release rate increased from 0.0020 to 0.0079. Colder regions in winter are more likely to experience a sense of crisis, increase the labor supply, and migrate to prevent potential future uncertainties.

From Table 6, we observe that the increase in daily temperature difference is conducive to the alleviation of the probability of personal vulnerability. As the standard poverty line increases, this effect becomes more obvious. The coefficient changed from 0.0017 to 0.0066. As the saying goes, “not cold or hot, grains are not growing (minor scene).” Meaning, the large temperature difference is conducive to the improvement of agricultural production, which has improved the living standard of individuals to a certain extent.

In summary, hotter summers, colder winters, and larger daily temperature differences were more conducive to reducing the vulnerability of individuals to poverty. 

## 5. Robustness Analysis

### 5.1. Vulnerable Probability Grouping

Although our research objectives were supported by the results of the benchmark regression, hotter summers and colder winters help reduce vulnerability to poverty. However, as described in Table 6, vulnerable groups and non-vulnerable groups may experience heterogeneous performance in the face of climate. Therefore, we have distinguished between vulnerable groups (fragility probability greater than 0.29), and non-vulnerable groups under different standards. Table 7 and Table 8 present the impact of extreme heat on vulnerable and non-vulnerable groups:

As shown in Table 7 and Table 8, we found that only high temperatures for non-vulnerable groups are beneficial for reducing vulnerability to poverty. However, high temperatures can increase individual vulnerability to poverty for vulnerable groups. This may be because vulnerable groups are more sensitive to high temperatures and more likely to return to poverty. The non-vulnerable group can use high-temperature characteristics of summer for their advantages to improve production, life, and their ability to prevent return to poverty.

As shown in Table 9 and Table 10, after being divided into vulnerable and non-vulnerable groups, the results of the heterogeneous grouping were consistent with the benchmark results. The colder the winter, the lower the vulnerability to poverty.

In summary, after preliminary grouping, we found that there was individual heterogeneity in the impact of summer on personal vulnerability to poverty. Is this heterogeneity because of a simple change in coefficient sign? Or is it a non-monotonic nonlinear transformation? In the following sections, we introduce a threshold regression model to examine whether an individual’s response to climate is related to their vulnerability to poverty.

### 5.2. Median Grouping of Altitude

The regression results are presented in Table 11. After grouping according to the median altitude of 41 county-level cities, the altitude above the median was set to 1, and otherwise was set to 0. In the table, (2), (4), (6), (8), and (10) are high-altitude areas, and (1), (3), (5), (7), and (9) are low-altitude areas. It was found that hotter summers are more beneficial to high-altitude areas because it is easier to reduce vulnerability to poverty in the region. Alternatively, hotter summers increase vulnerability to poverty for lower altitudes. The results can be interpreted more intuitively because plain and hilly areas are more likely to make people return to poverty under extreme summer heat. High-altitude areas can alleviate this high temperature pressure and reduce people’s vulnerability in summer. Similarly, Table 12 presents the results for the winter.

## 6. Mechanism Analysis

In this section, we introduce channel analysis to further explore our basic conclusions. Using “Maturity of Law”, we examined whether the subjective initiative of an individual improves poverty and vulnerability from the perspective of “man-made victory.” This issue was examined through the two channels of migrant workers and entrepreneurship. 

### 6.1. External Labor

The following Table 13 and Table 14 present the impact of extreme low temperature as well as heat on poverty vulnerability of migrated rural labors.

We consider an individual with a migrant worker at home to be 1 and an individual with no migrant worker at home to be 0. The results in Table 13 and Table 14 show that, whether it is hotter in summer or colder in winter, having migrant workers at home can reduce the probability of personal vulnerability to poverty. This is because migrant workers generate more income for the family and improve their living standards. 

### 6.2. Entrepreneurial Activity

We selected samples from individuals over 16 years of age. The results in Table 15 and Table 16 show that unlike individual migrant behaviors, individual entrepreneurial behaviors can significantly increase the vulnerability of individuals in high-temperature environments. Although this is not significant in low-temperature environments, the coefficient is still positive, which indicates that entrepreneurial behavior in any season increases the probability of individual vulnerability. This may be due to the large initial investment costs of starting a business, which reduces the current income of households and individuals. 

## 7. Threshold Inspection at Micro and Macro Levels

First, we confirmed that people with different levels of vulnerability are sensitive to temperature. Second, we verified that different levels of poverty and vulnerability differently affect individual poverty. Third, we examined how people with different levels of poverty respond to individual vulnerability to poverty. Finally, we combined all the sample individuals in each county to create a county-level city, calculated the average vulnerability to poverty of each county, and used it as the threshold variable. The highest temperature, lowest temperature, and maximum daily temperature difference were used as threshold variables. Based on this we determined threshold provinces, that is, provinces with the highest probability of causing poverty. 

### 7.1. Micro Subject Threshold Regression—Climate

Because we think that different vulnerability groups have different sensitivities to temperature in different groups, we used the cross-sectional threshold model for threshold returns to establish a single threshold model for vulnerability to poverty. The threshold variables are poverty vulnerability, which is explained by the maximum high temperature value, minimum low temperature value, and maximum daily temperature difference. The interpreted variables were the five standard poverty vulnerability lines, where X is the same as the previous control variables. We controlled for both county fixed effects and our model settings are as follows:$$Poor\_standard_{i}=\alpha_{1}\max_{i}\left(Poor\_standard_{i}\leq \gamma \right)+\alpha_{2}\max_{i}\left(Poor\_standard_{i}>\gamma \right)+\sum X_{i}+\varepsilon_{i}$$

#### 7.1.1. Highest Temperature

Table 17 shows, after using BS 300 times, that the F value was statistically significant, indicating that the test succeeded. As the subsequent self-sampling tests have succeeded, they will not be repeated here.

Table 18 shows the probability of poverty and vulnerability crossed the threshold, and that the hotter the summer, the more vulnerable the population became to the climate under different standards. This confirmed the results of our heterogeneous grouping.

#### 7.1.2. Lowest Temperatures

According to Table 19, colder winters are only beneficial for those with low vulnerability under different standards. As the probability of poverty and vulnerability crossed the threshold, the lower the winter temperature, the worse the situation for the high vulnerability group. Does this increase the probability of returning to poverty?

#### 7.1.3. Temperature Gap

As shown in Table 20, the temperature difference was the same as the minimum temperature. Once it crosses the threshold, it becomes disadvantageous for those who are extremely poor (those on the verge of vulnerability).

### 7.2. Micro Subject Threshold Regression-Poverty

On the one hand, we examined the extent to which people with different levels of poverty respond to individual poverty vulnerabilities. On the other hand, we verified that different levels of poverty vulnerability have different levels of impact on personal poverty. Our model settings are as follows:$$Poor\_standard_{i}=\alpha_{1}ln\_inc_{i}\left(Poor\_standard_{i}\leq \gamma \right)+\alpha_{2}ln\_inc_{i}(Poor\_standard_{i}>\gamma )+\sum X_{i}+\varepsilon_{i}$$
$$ln\_inc_{i}=\alpha_{1}Poor\_standard_{i}\left(\ln\_inc_{i}\leq \gamma \right)+\alpha_{2}Poor\_standard_{i}(\ln\_inc_{i}>\gamma )+\sum X_{i}+\varepsilon_{i}$$

#### 7.2.1. From Poverty to Poverty Vulnerability 

According to Table 21, the probability of returning to poverty decreases with an increase in income among the population with low vulnerability. In contrast, the probability of returning to poverty increases as income increases.

#### 7.2.2. From Poverty Vulnerability to Poverty 

According to Table 22, we considered a population whose income before the structural mutation point is poor and, conversely, non-poor. The results depict that for the poor, the greater the probability of returning to poverty and the greater the probability of poverty.

### 7.3. Macro Regional Threshold Regression 

#### 7.3.1. Highest Temperature

The following Table 23 shows the threshold estimations and confidence intervals as well as co-efficient changes. 

#### 7.3.2. Lowest Temperature

Table 24 shows that Shanxi, Shandong, Jiangsu, Hubei and Hunan provinces are more vulnerable to a return to poverty after the threshold. 

#### 7.3.3. Temperature Gap

Table 25 shows the results that Shandong, Shanxi Province is located after the threshold, which means it is easy to return to poverty.

## 8. Conclusions

The benchmark results demonstrated that extreme temperatures (hotter in the summer, colder in the winter, and greater day-to-day temperature gaps) help reduce vulnerability to poverty. The same conclusion was reached after using the median and average temperatures. After heterogeneous grouping, it was concluded that vulnerable groups are more likely to become poor in the face of extremely high temperatures, and non-vulnerable groups are shielded from returning to poverty. Colder winters are beneficial for both vulnerable and non-vulnerable groups. Higher altitudes are beneficial for reducing the probability of individuals returning to poverty, and migrant behaviors and entrepreneurship have opposite effects on alleviating the vulnerability to poverty. To further examine the sensitivity of individuals returning to poverty from extreme weather, it was found that different groups have different responses to climate despite a single threshold. This further substantiates the basic conclusions. Second, there is a linkage effect between poverty vulnerability and poverty; that is, poorer people will become poorer with the increase in poverty vulnerability. Further, in the context of higher income, the higher the probability of returning to poverty, the higher the vulnerability. Finally, different provinces should adopt targeted strategies to alleviate the threat to poverty due to extreme weather. Therefore, special attention should be paid to climate-sensitive vulnerable individuals. Both the dimensions of poverty and vulnerability to poverty should be considered while formulating relevant policy measures. In addition, different provinces should adopt different strategies to deal with climate problems. Further, an early risk warning mechanism should be established to guide farmers’ entrepreneurial behavior. For plain hilly areas in hot summers, it is recommended to develop befitting support measures.

However, limited by data, our empirical research did not achieve some of the goals that we set before investigation. First, due to the lack of panel data, especially those extending to the year 2020, we cannot completely observe the relevant characteristics of poverty caused by climate change in the process of one of the largest anti-poverty campaigns in human history. Second, due to lack of data on fuel, electricity, and other domestic energy consumption, we cannot incorporate excessive energy consumption caused by climate change into the impact on poverty or quality of life. Third, this paper neglects the impact on urban poor residents. Temperature should also have a strong impact on urban residents, especially among suburban residents and immigrants, and its mechanism is likely to be quite different from that of rural residents. We will address these shortcomings in the future investigations. Finally, the extreme temperature, especially the temperature rise, inevitably leads to the need to use energy to adjust the temperature of life and production. Because the poor are more sensitive to the extra expenditure caused by energy use, which indirectly increases the pressure on the poor, the impact of temperature rise on the energy sector may also have an extra impact on poverty. However, due to the limitation of the length of this article, we put this discussion in the future research goal.

## Figures and Tables

**Figure 1 ijerph-19-03328-f001:**
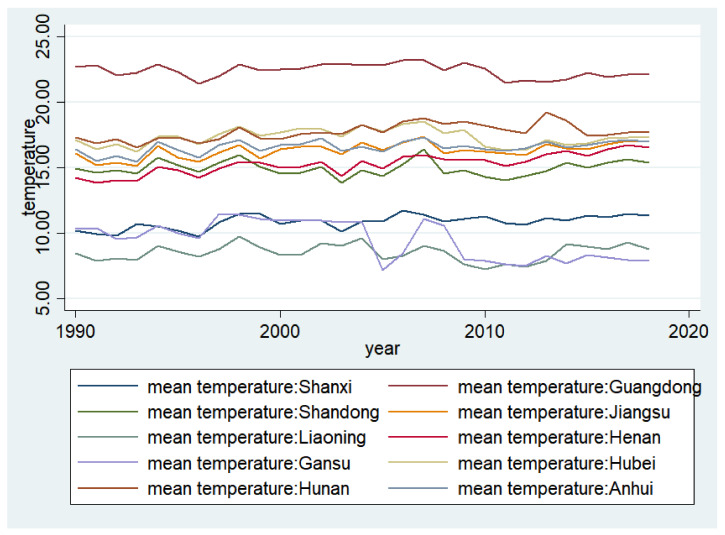
Average temperature by province.

**Table 1 ijerph-19-03328-t001:** Explanation and description of related variables.

Variables	Explanation	Threshold
*Poor_$1*	World Bank $1 Standard	2270
*Poor_low*	30% of the median per capita disposable income of the National Bureau of Statistics in 2013	2825.64
*Poor_$1.5*	World Bank $1.5 Standard	3441
*Poor_high*	50% of the median per capita disposable income of the National Bureau of Statistics in 2013	4521.04
*Poor_$2*	World Bank $2 Standard	4587

**Table 2 ijerph-19-03328-t002:** Explanation and description of related variables.

Variable Name	Meaning	Mean	Standard Error	Min	Max	Observation
Dependent Variable						
*Poor_$1*	World Bank $1 Standard	0.023	0.076	0	1	8416
*Poor_low*	30% of the median per capita disposable income of the National Bureau of Statistics in 2013	0.031	0.091	0	1	8416
*Poor_$1.5*	World Bank $1.5 Standard	0.039	0.106	0	1	8416
*Poor_high*	50% of the median per capita disposable income of the National Bureau of Statistics in 2013	0.055	0.131	0	1	8416
*Poor_$2*	World Bank $2 Standard	0.056	0.132	0	1	8416
Independent variable						
*Max*	Max high temperature	36.879	3.748	18	41	13,218
*Min*	Minimum Low temperature	−6.681	7.077	−29	5	13,218
*Median_max*	High-temperature median	22.328	3.875	7	29	13,218
*Median_min*	Low temperature median	11.942	4.762	0	20	13,218
*Mean_max*	High temperature average	21.370	3.701	9.858	28.159	13,218
*Mean_min*	Low temperature mean value	11.725	4.427	0.315	18.918	13,218
*Maxgap*	Annual max of daily temperature gap	18.9471	2.505	14	26	13,218
Control variable						
*ln*_*income*	Log of personal current income	10.560	0.741	6.397	13.370	13,203
*Ethnic*	Ethnic	1.321	1.428	1	8	13,217
*Party*	Party	2.910	0.413	1	3	13,138
*Administrator*	Rural administrator	5.930	0.502	1	6	13,141
*Resident*	Hukou	1.153	0.534	1	4	13,215
*Consumption*	Min consumption level	25,207.08	16,381.41	900	400,000	13,093
*Education*	Educational level	2.884	1.360	1	9	12,731
*Health*	Health condition	1.939	0.873	1	5	13,205
*Medicare*	Medical security	3.946	0.674	1	7	13,218
*Ltd*	Longitude	114.229	4.695	101.971	122.26	13,218
*Latd*	Latitude	31.856	4.780	22.692	41.23	13,218
Other						
*Alttud*	Altitude	203.496	350.056	7	1496	13,218

**Table 3 ijerph-19-03328-t003:** Group descriptive statistical analysis.

Grouping	(1)	(2)	(3)	(4)	(5)
	*Poor_$1*	*Poor_Low*	*Poor_$1.5*	*Poor_High*	*Poor_$2*
High altitude	3900	3976	4063	4227	4246
Low altitude	1038	1051	1070	1104	1110
Max temp high	1757	1758	1774	1790	1794
Max temp low	3187	3269	3359	3514	3562
Min temp high	2012	2020	2043	2063	2069
Min temp low	2926	3007	3090	3268	3287
Gap small	1732	1759	1790	1859	1864
Gap big	3206	3268	3343	3472	3492

**Table 4 ijerph-19-03328-t004:** Impact of extreme heat on poverty vulnerability.

	(1)	(2)	(3)	(4)	(5)
	*Poor_$1*	*Poor_Low*	*Poor_$1.5*	*Poor_High*	*Poor_$2*
*Max*	−0.0131 ***	−0.0251 ***	−0.0343 ***	−0.0508 ***	−0.0519 ***
	(0.0025)	(0.0028)	(0.0030)	(0.0033)	(0.0034)
Constant	−0.8162 ***	−1.4421 ***	−2.1546 ***	−3.5099 ***	−3.5947 ***
	(0.0898)	(0.0934)	(0.0996)	(0.1144)	(0.1152)
fixed effect	YES	YES	YES	YES	YES
	8407	8407	8407	8407	8407
adj. R^2^	0.557	0.599	0.636	0.678	0.680

Note: Standard errors in parentheses. *** *p* < 0.01.

**Table 5 ijerph-19-03328-t005:** Impact of extreme low temperatures on poverty vulnerability.

	(1)	(2)	(3)	(4)	(5)
	*Poor_$1*	*Poor_Low*	*Poor_$1.5*	*Poor_High*	*Poor_$2*
*Min*	0.0020 ***	0.0038 ***	0.0052 ***	0.0077 ***	0.0079 ***
	(0.0004)	(0.0004)	(0.0005)	(0.0005)	(0.0005)
Constant	−0.7938 ***	−1.3991 ***	−2.0959 ***	−3.4228 ***	−3.5059 ***
	(0.0879)	(0.0914)	(0.0975)	(0.1117)	(0.1125)
fixed effect	YES	YES	YES	YES	YES
N	8407	8407	8407	8407	8407
adj. R^2^	0.557	0.599	0.636	0.678	0.680

Note: Standard errors in parentheses. *** *p* < 0.01.

**Table 6 ijerph-19-03328-t006:** Impact of daily temperature differences on poverty vulnerability.

	(1)	(2)	(3)	(4)	(5)
	*Poor_$1*	*Poor_Low*	*Poor_$1.5*	*Poor_High*	*Poor_$2*
*Max*gap	−0.0017 ***	−0.0032 ***	−0.0044 ***	−0.0065 ***	−0.0066 ***
	(0.0003)	(0.0004)	(0.0004)	(0.0004)	(0.0004)
Constant	−0.5240 ***	−0.8813 ***	−1.3884 ***	−2.3738 ***	−2.4354 ***
	(0.0813)	(0.0876)	(0.0941)	(0.1026)	(0.1029)
Fixed effect	YES	YES	YES	YES	YES
N	8407	8407	8407	8407	8407
adj. R^2^	0.557	0.599	0.636	0.678	0.680

Note: Standard errors in parentheses. *** *p* < 0.01.

**Table 7 ijerph-19-03328-t007:** Impact of extreme heat on groups with a probability of poverty greater than 0.29.

	(1)	(2)	(3)	(4)	(5)
	*Poor_$1*	*Poor_Low*	*Poor_$1.5*	*Poor_High*	*Poor_$2*
*Max*	0.1169 ***	0.0690 ***	0.0141 ***	0.2239 ***	0.2036 ***
	(0.0100)	(0.0121)	(0.0042)	(0.0428)	(0.0408)
Constant	−1.6791	4.2038 ***	4.3982 ***	5.4792 ***	5.0855 ***
	(0.9759)	(0.9494)	(0.6455)	(1.0423)	(1.0160)
Fixed effect	YES	YES	YES	YES	YES
N	136	225	331	529	554
adj. R^2^	0.663	0.692	0.701	0.754	0.764

Note: Standard errors in parentheses. *** *p* < 0.01.

**Table 8 ijerph-19-03328-t008:** Impact of extreme heat on groups with a probability of poverty less than 0.29.

	(1)	(2)	(3)	(4)	(5)
	*Poor_$1*	*Poor_Low*	*Poor_$1.5*	*Poor_High*	*Poor_$2*
*Max*	−0.0110 ***	−0.0180 ***	−0.0203 ***	−0.0291 ***	−0.0299 ***
	(0.0013)	(0.0015)	(0.0018)	(0.0022)	(0.0022)
Constant	−0.8470 ***	−1.4106 ***	−1.5676 ***	−2.2732 ***	−2.3382 ***
	(0.0484)	(0.0451)	(0.0664)	(0.0847)	(0.0862)
Fixed effect	YES	YES	YES	YES	YES
N	8271	8182	8076	7878	7853
adj. R^2^	0.528	0.562	0.584	0.604	0.602

Note: Standard errors in parentheses. *** *p* < 0.01.

**Table 9 ijerph-19-03328-t009:** Effect of extreme low temperature on groups with a probability of poverty greater than 0.29.

	(1)	(2)	(3)	(4)	(5)
	*Poor_$1*	*Poor_Low*	*Poor_$1.5*	*Poor_High*	*Poor_$2*
*Min*	0.1313 ***	0.0775 ***	0.0158 ***	0.3186 ***	0.2896 ***
	(0.0113)	(0.0136)	(0.0047)	(0.0609)	(0.0581)
Constant	−19.4937 ***	−6.3187 ***	2.2496 ***	−36.9545 ***	−33.4884 ***
	(1.2895)	(1.3986)	(0.7917)	(7.1649)	(6.8138)
Fixed effect	YES	YES	YES	YES	YES
N	136	225	331	529	554
adj. R^2^	0.663	0.692	0.701	0.754	0.764

Note: Standard errors in parentheses. *** *p* < 0.01.

**Table 10 ijerph-19-03328-t010:** Impact of extreme low temperature on groups with a probability of poverty greater than 0.29.

	(1)	(2)	(3)	(4)	(5)
	*Poor_$1*	*Poor_Low*	*Poor_$1.5*	*Poor_High*	*Poor_$2*
*Min*	0.0017 ***	0.0027 ***	0.0031 ***	0.0044 ***	0.0046 ***
	(0.0002)	(0.0002)	(0.0003)	(0.0003)	(0.0003)
Constant	−0.8282 ***	−1.3798 ***	−1.5329 ***	−2.2233 ***	−2.2870 ***
	(0.0464)	(0.0429)	(0.0640)	(0.0813)	(0.0828)
Fixed effect	YES	YES	YES	YES	YES
N	8271	8182	8076	7878	7853
adj. R^2^	0.528	0.562	0.584	0.604	0.602

Note: Standard errors in parentheses. *** *p* < 0.01.

**Table 11 ijerph-19-03328-t011:** Impact of extreme heat on poverty vulnerability—altitude.

	(1)	(2)	(3)	(4)	(5)	(6)	(7)	(8)	(9)	(10)
	$1	$1	Low	Low	$1.5	$1.5	High	High	$2	$2
*Max*	0.0102 ***	−0.0163 ***	0.0123 ***	−0.0292 ***	0.0145 ***	−0.0392 ***	0.0178 ***	−0.0571 ***	0.0180 ***	−0.0582 ***
	(0.002)	(0.002)	(0.002)	(0.003)	(0.002)	(0.003)	(0.003)	(0.004)	(0.003)	(0.004)
Constant	3.0572 ***	−0.8833 ***	3.6511 ***	−1.5401 ***	4.1917 ***	−2.2792 ***	4.8723 ***	−3.6711 ***	4.9044 ***	−3.7582 ***
	(0.368)	(0.122)	(0.427)	(0.126)	(0.472)	(0.134)	(0.507)	(0.153)	(0.508)	(0.154)
Fixed effect	YES	YES	YES	YES	YES	YES	YES	YES	YES	YES
N	1712	6695	1712	6695	1712	6695	1712	6695	1712	6695
adj. R^2^	0.574	0.555	0.613	0.598	0.643	0.637	0.681	0.680	0.684	0.682

Note: Standard errors in parentheses. *** *p* < 0.01.

**Table 12 ijerph-19-03328-t012:** Impact of extreme low temperature on poverty vulnerability—altitude.

	(1)	(2)	(3)	(4)	(5)	(6)	(7)	(8)	(9)	(10)
	$1	$1	Low	Low	$1.5	$1.5	High	High	$2	$2
*Max*	0.0116 ***	0.0025 ***	0.0140 ***	0.0044 ***	0.0164 ***	0.0060 ***	0.0202 ***	0.0087 ***	0.0204 ***	0.0089 ***
	(0.002)	(0.000)	(0.003)	(0.000)	(0.003)	(0.001)	(0.003)	(0.001)	(0.003)	(0.00)
Constant	2.8032 ***	−0.8554 ***	3.3443 ***	−1.4901 ***	3.8304 ***	−2.2121 ***	4.4278 ***	−3.5734 ***	4.4554 ***	−3.6585 ***
	(0.417)	(0.120)	(0.482)	(0.124)	(0.530)	(0.132)	(0.569)	(0.151)	(0.570)	(0.151)
Fixed effect	YES	YES	YES	YES	YES	YES	YES	YES	YES	YES
N	1712	6695	1712	6695	1712	6695	1712	6695	1712	6695
adj. R^2^	0.574	0.555	0.613	0.598	0.643	0.637	0.681	0.680	0.684	0.682

Note: Standard errors in parentheses. *** *p* < 0.01.

**Table 13 ijerph-19-03328-t013:** Impact of extreme low temperature on poverty vulnerability—migrant workers.

	(1)	(2)	(3)	(4)	(5)
	*Poor_$1*	*Poor_Low*	*Poor_$1.5*	*Poor_$1.5*	*Poor_$2*
*Min*	0.0025 ***	0.0044 ***	0.0057 ***	0.0057 ***	0.0081 ***
	(0.0005)	(0.0005)	(0.0006)	(0.0006)	(0.0006)
*External*	−0.0043 **	−0.0052 ***	−0.0061 ***	−0.0061 ***	−0.0073 ***
	(0.0016)	(0.0019)	(0.0022)	(0.0022)	(0.0026)
*Min* * *External*	−0.0006 **	−0.0008 **	−0.0009 **	−0.0009 **	−0.0010 **
	(0.0003)	(0.0003)	(0.0004)	(0.0004)	(0.0004)
Constant	−1.2515 ***	−1.9103 ***	−2.6184 ***	−2.6184 ***	−3.9686 ***
	(0.0954)	(0.0972)	(0.1006)	(0.1006)	(0.1130)
Fixed effect	YES	YES	YES	YES	YES
N	6803	6803	6803	6803	6803
adj. R^2^	0.566	0.609	0.646	0.646	0.689

Note: Standard errors in parentheses. ** *p* < 0.05, *** *p* < 0.01.

**Table 14 ijerph-19-03328-t014:** Impact of extreme heat on poverty vulnerability—migrant workers.

	(1)	(2)	(3)	(4)	(5)
	*Poor_$1*	*Poor_Low*	*Poor_$1.5*	*Poor_$1.5*	*Poor_$2*
*Max*	−0.0144 ***	−0.0267 ***	−0.0351 ***	−0.0351 ***	−0.0505 ***
	(0.0036)	(0.0039)	(0.0041)	(0.0041)	(0.0044)
*External*	0.0327 *	0.0351	0.0399	0.0399	0.0430
	(0.0187)	(0.0221)	(0.0248)	(0.0248)	(0.0272)
*Max* * *External*	−0.0009 *	−0.0010	−0.0011 *	−0.0011 *	−0.0012
	(0.0005)	(0.0006)	(0.0006)	(0.0006)	(0.0007)
Constant	−1.2534 ***	−1.9289 ***	−2.6474 ***	−2.6474 ***	−4.0198 ***
	(0.1009)	(0.1016)	(0.1033)	(0.1033)	(0.1131)
Fixed effect	YES	YES	YES	YES	YES
N	6803	6803	6803	6803	6803
adj. R^2^	0.566	0.609	0.646	0.646	0.688

Note: Standard errors in parentheses. * *p* < 0.1, *** *p* < 0.01.

**Table 15 ijerph-19-03328-t015:** Impact of extreme low temperature on poverty vulnerability—entrepreneurship.

	(1)	(2)	(3)	(4)	(5)
	*Poor_$1*	*Poor_Low*	*Poor_$1.5*	*Poor_$1.5*	*Poor_$2*
*Min*	0.0021 ***	0.0040 ***	0.0055 ***	0.0081 ***	0.0083 ***
	(0.0005)	(0.0005)	(0.0005)	(0.0006)	(0.0006)
*Entrepre*	0.0039 *	0.0051 **	0.0065 **	0.0091 **	0.0093 **
	(0.0021)	(0.0025)	(0.0029)	(0.0038)	(0.0038)
*Min* * *Entrepre*	0.0005 **	0.0008 ***	0.0011 ***	0.0016 ***	0.0016 ***
	(0.0003)	(0.0003)	(0.0003)	(0.0003)	(0.0004)
Constant	−0.8174 ***	−1.4404 ***	−2.1549 ***	−3.5060 ***	−3.5902 ***
	(0.1006)	(0.1020)	(0.1054)	(0.1158)	(0.1165)
Fixed effect	YES	YES	YES	YES	YES
N	8407	8407	8407	8407	8407
adj. R^2^	0.557	0.600	0.636	0.678	0.681

Note: Standard errors in parentheses. * *p* < 0.1, ** *p* < 0.05, *** *p* < 0.01.

**Table 16 ijerph-19-03328-t016:** Impact of extreme heat on poverty vulnerability—entrepreneurship.

	(1)	(2)	(3)	(4)	(5)
	*Poor_$1*	*Poor_Low*	*Poor_$1.5*	*Poor_$1.5*	*Poor_$2*
*Max*	−0.0143 ***	−0.0272 ***	−0.0373 ***	−0.0550 ***	−0.0561 ***
	(0.0031)	(0.0034)	(0.0037)	(0.0042)	(0.0043)
*Entrepre*	−0.0266	−0.0410	−0.0543	−0.0747	−0.0758
	(0.0212)	(0.0270)	(0.0337)	(0.0467)	(0.0475)
*Max* * *Entrepre*	0.0007	0.0011	0.0014	0.0020	0.0020
	(0.0006)	(0.0007)	(0.0009)	(0.0012)	(0.0013)
Constant	−0.8364 ***	−1.4787 ***	−2.2062 ***	−3.5814 ***	−3.6672 ***
	(0.1011)	(0.1032)	(0.1084)	(0.1237)	(0.1246)
Fixed effect	YES	YES	YES	YES	YES
N	8407	8407	8407	8407	8407
adj. R^2^	0.557	0.599	0.636	0.678	0.680

Note: Standard errors in parentheses. *** *p* < 0.01.

**Table 17 ijerph-19-03328-t017:** Threshold effect of self-sampling test.

Threshold Test Type	Standard	F Metrology	*p* Value	Number of BS	Critical Value		
					1%	5%	10%
Single	*Poor_$1*	9511.174 ***	0.000	300	6.448	4.314	3.153
Single	*Poor_low*	9537.620 ***	0.000	300	6.134	3.721	2.492
Single	*Poor_$1.5*	1.0 × 10^4^ ***	0.000	300	7.009	4.152	3.153
Single	*Poor_high*	1.1 × 10^4^ ***	0.000	300	8.835	3.734	2.273
Single	*Poor_$2*	1.1 × 10^4^ ***	0.000	300	7.552	3.689	2.494

Note: *p*-value and critical value are the results of repeated sampling by the self-sampling method; *** *p* < 0.01.

**Table 18 ijerph-19-03328-t018:** Threshold estimates and confidence intervals and coefficient changes.

Standard	Threshold	95% Interval	Lower Coefficient	Higher Coefficient
*Poor_$1*	0.239	[0.239, 0.239]	0.00307 *** (8.35)	0.0106 *** (28.48)
*Poor_low*	0.215	[0.215, 0.215]	0.00407 *** (9.69)	0.0112 *** (26.45)
*Poor_$1.5*	0.212	[0.212, 0.212]	0.00426 *** (9.25)	0.0154 *** (24.01)
*Poor_high*	0.267	[0.267, 0.267]	0.00755 *** (14.34)	0.0152 *** (28.83)
*Poor_$2*	0.271	[0.246, 0.271]	0.00775 *** (14.60)	0.0154 *** (29.09)

Standard errors in parentheses. *** *p* < 0.01.

**Table 19 ijerph-19-03328-t019:** Threshold estimates, confidence intervals, and coefficient changes.

Standard	Threshold	95% Interval	Lower Coefficient	Higher Coefficient
*Poor_$1*	0.239	[0.239, 0.239]	−0.00357 *** (−7.34)	−0.0192 *** (−37.14)
*Poor_low*	0.215	[0.200, 0.215]	−0.00460 *** (−8.25)	−0.0195 *** (−33.75)
*Poor_$1.5*	0.194	[0.194, 0.194]	−0.00459 *** (−7.38)	−0.0183 *** (−29.26)
*Poor_high*	0.235	[0.235, 0.267]	0.00786 *** (−10.85)	−0.0232 *** (−31.93)
*Poor_$2*	0.243	[0.243, 0.246]	−0.0081 *** (−11.04)	−0.0236 *** (−32.19)

Standard errors in parentheses. *** *p* < 0.01.

**Table 20 ijerph-19-03328-t020:** Threshold estimates, confidence intervals, and coefficient changes.

Standard	Threshold	95% Interval	Lower Coefficient	Higher Coefficient
*Poor_$1*	0.135	[0.135, 0.135]	0.0000744 (0.42)	0.0115 *** (55.43)
*Poor_low*	0.168	[0.168, 0.168]	−0.000447 ** (−2.20)	0.0124 *** (54.28)
*Poor_$1.5*	0.173	[0.173, 0.173]	−0.000321 *** (−1.42)	0.0134 *** (53.79)
*Poor_high*	0.188	[0.188, 0.188]	−0.00107 *** (−4.02)	0.0137 *** (48.61)
*Poor_$2*	0.188	[0.188, 0.188]	−0.00101 *** (−3.77)	0.0138 *** (48.86)

Standard errors in parentheses. ** *p* < 0.05, *** *p* < 0.01.

**Table 21 ijerph-19-03328-t021:** Threshold estimates, confidence intervals, and coefficient changes.

Standard	Threshold	95% Interval	Lower Coefficient	Higher Coefficient
*Poor_$1*	0.123	[0.123, 0.123]	−0.00352 *** (−9.01)	0.0147 *** (36.92)
*Poor_low*	0.121	[0.121, 0.121]	−0.00345 *** (−9.01)	0.0145 *** (37.18)
*Poor_$1.5*	0.090	[0.090, 0.090]	−0.00299 *** (−11.24)	0.0101 *** (37.53)
*Poor_high*	0.072	[0.072, 0.072]	−0.00198 *** (−9.15)	0.00931 *** (42.36)
*Poor_$2*	0.055	[0.055, 0.055]	−0.00156 *** (−9.20)	0.00733 *** (42.53)

Standard errors in parentheses. *** *p* < 0.01.

**Table 22 ijerph-19-03328-t022:** Threshold estimates, confidence intervals, and coefficient changes.

Standard	Threshold	95% Interval	Lower Coefficient	Higher Coefficient
*Poor_$1*	10.143	[10.127, 10.166]	−9.299 *** (−32.46)	3.515 *** (16.85)
*Poor_low*	10.143	[10.127, 10.166]	−7.512 *** (−33.80)	2.856 *** (17.55)
*Poor_$1.5*	10.143	[10.127, 10.166]	−6.237 *** (−35.09)	2.386 *** (18.21)
*Poor_high*	10.143	[10.127, 10.166]	−4.888 *** (−37.05)	1.865 *** (18.97)
*Poor_$2*	10.143	[10.127, 10.166]	−4.828 *** (−37.16)	1.842 *** (19.01)

Standard errors in parentheses. *** *p* < 0.01.

**Table 23 ijerph-19-03328-t023:** Threshold estimates, confidence intervals, and coefficient changes.

Standard	Threshold	95% Interval	Lower Coefficient	Higher Coefficient
*Poor_$1*	0.055	[0.055, 0.055]	−0.000352 *** (12.07)	0.00296 *** (95.74)
*Poor_low*	0.072	[0.072, 0.072]	0.000490 *** (13.07)	0.00380 *** (95.54)
*Poor_$1.5*	0.090	[0.090, 0.090]	0.000778 *** (16.90)	0.00464 *** (95.31)
*Poor_high*	0.121	[0.121, 0.121]	0.00103 *** (15.16)	0.00627 *** (87.27)
*Poor_$2*	0.123	[0.123, 0.123]	0.00105 *** (15.19)	0.00636 *** (86.86)

Standard errors in parentheses. *** *p* < 0.01.

**Table 24 ijerph-19-03328-t024:** Threshold estimates, confidence intervals, and coefficient changes.

Standard	Threshold	95% Interval	Lower Coefficient	Higher Coefficient
*Poor_$1*	0.013	[0.012, 0.013]	0.000894 *** (8.76)	−0.00441 *** (−104.69)
*Poor_low*	0.019	[0.019, 0.019]	0.00102 *** (19.77)	−0.00580 *** (−116.99)
*Poor_$1.5*	0.029	[0.029, 0.029]	0.000966 *** (16.13)	−0.00750 *** (−130.70)
*Poor_high*	0.047	[0.047, 0.047]	0.000655 *** (8.76)	−0.0105 *** (−146.33)
*Poor_$2*	0.049	[0.049, 0.049]	0.000629 *** (8.31)	−0.0107 *** (−146.93)

Standard errors in parentheses. *** *p* < 0.01.

**Table 25 ijerph-19-03328-t025:** Threshold estimates, confidence intervals, and coefficient changes.

Standard	Threshold	95% Interval	Lower Coefficient	Higher Coefficient
*Poor_$1*	0.055	[0.055, 0.055]	−0.000900 *** (−17.28)	0.00558 *** (65.90)
*Poor_low*	0.072	[0.072, 0.072]	−0.000839 *** (−13.06)	0.00499 *** (72.96)
*Poor_$1.5*	0.090	[0.090, 0.090]	−0.00127 *** (−16.75)	0.00558 *** (70.12)
*Poor_high*	0.121	[0.121, 0.121]	−0.000231 ** (−2.20)	0.00499 *** (83.97)
*Poor_$2*	0.123	[0.123, 0.123]	−0.000262 ** (−2.53)	0.00921 *** (83.84)

Standard errors in parentheses. ** *p* < 0.05, *** *p* < 0.01.

## Data Availability

Chinese Household Income Project (CHIP) (http://www.ciidbnu.org/chip/index.asp, accessed on 5 October 2019); Hourly data from surface meteorological stations in China (https://data.cma.cn/en, accessed on 5 October 2019); Chinese Research Data Services (https://www.cnrds.com/, accessed on 5 October 2019).

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
