# Peer review of "Impact of Climate Change on Rural Poverty Vulnerability from an Income Source Perspective: A Study Based on CHIPS2013 and County-Level Temperature Data in China"

_ijerph, 2022, doi:10.3390/ijerph19063328_

Round 1
Reviewer 1 Report
Manuscript ID: ijerph-1564109
Authors focus measuring the climate impact on poverty vulnerability.
In my opinion the paper is very interesting. It is very easy to follow and it is very well written.
I have got some minor comments:
- Authors follow the VEP method and they explain it. However, they should add some bibliography on it.
- The analyze poverty for the variables poor_2270, poor_low, poor_3441, poor_high, and poor_ But they do not define them. I cannot know what these indices are measuring. I suppose that they are energy poverty indicators but they should define them and their differences.
Author Response
Dear Reviewer,
Thank you for your valuable comments and direction to improve the paper. Please find our responses at the attached file. Please also read the clean version of the paper where we made thorough changes and improvement.
Thank you again for your supports.
regards,
Muhammad et al.

Reviewer 2 Report
The paper analyzes the impact of climate on vulnerability to individual poverty in rural China. The main results are: (1) extreme temperatures (hotter summers, colder winters, and greater day-to-15 day temperature gaps) reduce vulnerability to poverty; (2) there is an association between poverty vulnerability and poverty; 17 that is, poorer people will become poorer with an increase in poverty vulnerability.
1.The paper is well structured and written.
2.The title of the article is clear and adequate.
3.The abstract is clear, it presents the object of research, the content, and the results.
4.The introduction states the objectives of the paper.
- The methodology seems sound.
Major concerns
You should explain why China is different to other countries and why is it worth studying.
In the introduction and also in the literature review you should discuss also about the impact of temperature increase on energy sector that has an additional impact on poverty (see additional readings).
In the introduction you should highligh the main results and the contribution to the extant literature.
In the conclusions section, the authors should stress the limits of their empirical research.
Also, the authors should elaborate more on their contributions.
In conclusion, I would like to thank the authors for a very interesting, unique, and potentially important paper. Hope these comments and suggestions can help further their study. I consider that the paper can bring a significant contribution to the extant literature once the above-mentioned recommendations are taken into account.
Readings
The impact of temperature increase on firm profitability. Empirical evidence from the European energy and gas sectors, Applied Energy, 295, 117051, doi: 10.1016/j.apenergy.2021.117051
Author Response

(The authors gave the same response as above.)

Reviewer 3 Report
Congratulations to authors for an extremely interesting and valuable paper. Please take into consideration some minor suggestions:
- Define more clearly the fragile and non-fragile groups
- Better explain Hypothesis 4: ’Vulnerability associated with different thresholds caused by changes 207 in temperature and temperature gap, induces different states in different provinces’. What kind of states?
- Briefly outline the limitations of the research and future steps
- If possible detail the responsibility and main stakeholders involved for each policy recommendation
- Finally, please use the MDPI format for referring the bibliography in the paper
Author Response

(The authors gave the same response as above.)

Reviewer 4 Report
Thank you very much for the opportunity to read this text. The authors raise important social issues. It is one of the few works that combines aspects of climate change and the material situation of rural residents. Therefore, the text is all the more interesting. I think that in the title it should be noted that the work refers to China. My other comments:
1. „Introduction” section- it is a pity that the importance of the problem was not emphasized more in the introduction. It is not until the text in lines 58-67 that the relationship between climate change and rural impoverishment is indicated. In my opinion Fig 1 and 2 can be omitted. Instead, the need for this research should be better explained.
2. „Materials and methods” - literature review should not be presented in this part of the paper. It seems to me that it should be either an element of Introduction or precede the methodological chapter. The aim of the study was not to conduct a critical review of the literature.
3. A similar remark concerns hypotheses, which should be presented before the „Materials and methods” section. I think that hypothesis 1 is unnecessary. The purpose of the paper is not to analyze variable climate determinants but to determine the relationship between climate change and rural poverty levels, at least that is what the title of the paper implies.
4. The text in line 237 needs clarification, were the samples drawn only from rural households? It seems that including urban households may bias the results.
5. In line 245, the authors use the abbreviation of the VEP method. There should be a full name and a few words explaining what the method is and who developed it. I suggest moving the text of lines 349-387 to this place
6. There should be more explanation of what was the source of the data indicated in lines 276-280. How were these variables determined?
7. I believe that the text from lines 429-432 can be omitted.
8. There is a lack of discussion, limitations to the study, and it is a pity that recommendations for further research in this area are not included
Author Response

(The authors gave the same response as above.)

Round 2
Reviewer 2 Report
Dear Editors,
In my opinion, the authors didn’t take into consideration my recommendations. I consider that the paper can not be published in this form.
Best regards
Author Response
Dear Reviewer,
Thank you for your valuable comments. We have added the increase of temperature and firms profitability issue in the literature and updated the introduction, discussion, conclusion, and future research sections. We have also cited the reference you proposed.
Thank you again for your supports.

Reviewer 4 Report
Thank you for considering my comments
Author Response
Dear Reviewer,
Thank you for your comments and guidance to improve this paper. we have further improved the paper. please the clean version of the paper to see the changes we made.
Thank you.
